# Prognostic Implications of Maintaining the Target Thyroid-Stimulating Hormone Status Based on the 2015 American Thyroid Association Guidelines in Patients with Low-Risk Papillary Thyroid Carcinoma after Lobectomy: A 5-Year Landmark Analysis

**DOI:** 10.3390/cancers16193253

**Published:** 2024-09-24

**Authors:** Ye Won Jeon, Young Jin Suh, Seung Taek Lim

**Affiliations:** Division of Breast and Thyroid Surgical Oncology, Department of Surgery, St. Vincent’s Hospital, College of Medicine, The Catholic University of Korea, Suwon 16247, Republic of Korea; yeswon80@naver.com (Y.W.J.); yjsuh@catholic.ac.kr (Y.J.S.)

**Keywords:** papillary thyroid carcinoma, low-risk, lobectomy, thyroid stimulating hormone, landmark analysis, recurrence

## Abstract

**Simple Summary:**

Thyroid cancer is very common, with approximately 580,000 cases reported worldwide in 2020. Proper evaluation and management are necessary to ensure maximum patient care. It is important to consider the entire thyroid stimulating hormone (TSH) test results rather than the average or partial TSH test results to accurately evaluate the TSH status of patients with low-risk papillary thyroid carcinoma (PTC) after lobectomy. In patients with low-risk PTC after lobectomy, recurrence-free survival can be worse if the TSH status of patients, despite fulfilling the 2015 American Thyroid Association guideline recommendations, is maintained below a certain percentage among the total number of TSH tests during the postoperative follow-up period. Clinicians should make deliberate efforts to maintain the postoperative TSH status of patients with low-risk PTC who underwent lobectomy within an appropriate level during the postoperative follow-up period using thorough examinations.

**Abstract:**

**Background**: The 2015 American Thyroid Association guidelines recommend the maintenance of serum thyroid stimulating hormone (TSH) levels ≤2 mIU/L in patients with low-risk papillary thyroid carcinoma (PTC) who underwent lobectomy; however, the evidence is insufficient. We investigated the association between maintaining the TSH status at ≤2 mIU/L and tumor recurrence in patients with low-risk PTC who underwent lobectomy through a 5-year landmark analysis. **Methods**: Between 2010 and 2016, 662 patients with low-risk PTC were included. The postoperative TSH status was determined using the ‘TSH > 2 ratio’, which was calculated using the TSH test results during the 5-year follow-up. The optimal cutoff value of ‘TSH > 2 ratio’ for tumor recurrence was determined using a receiver operating characteristic curve analysis. Recurrence-free survival (RFS) was compared between the groups using Kaplan–Meier and Cox proportional hazard regression analyses. **Results**: Patients with ‘TSH > 2 ratio’ > 0.1833 (*n* = 498) had a worse RFS outcome compared to patients with ‘TSH > 2 ratio’ ≤ 0.1833 (*n* = 164; *p* < 0.001). ‘TSH > 2 ratio’ > 0.1833 was a significant risk factor for tumor recurrence after the 5-year landmark (hazard ratio: 4.795, 95% confidence interval: 2.102–10.937, *p* < 0.001). **Conclusions**: Maintaining TSH levels ≤ 2 mIU/L below a certain percentage among the total TSH tests during the 5-year follow-up period has a negative impact on tumor recurrence.

## 1. Introduction

Thyroid cancer is a common endocrine malignancy, with approximately 580,000 cases reported worldwide in 2020, and it ranks ninth in terms of incidence among all cancers [1]. Papillary thyroid cancer (PTC) is the most common subtype of thyroid cancer, accounting for more than 90% of all differentiated thyroid cancer (DTC) [2]. Due to recent developments in diagnostic equipment and the implementation of early screening programs, a large proportion of thyroid cancers are diagnosed at an early stage [3].

Most patients experience long-term survival if appropriate surgical treatment is conducted in the early stage. Disease-specific mortality is low at less than 2% in 10 years [4,5,6]. Therefore, to maintain a good postoperative quality of life during the long-term survival period for patients with early-stage thyroid cancer, it is important to maintain a good balance between therapeutic intervention and patient convenience [7].

Thyroid-stimulating hormone (TSH) suppression using thyroid hormone supplementation is generally recommended after surgery for PTC. This treatment strategy is based on the theory that TSH plays a role in promoting the proliferation of thyroid cancer cells; hence, suppressing TSH to a low level can repress cancer cell proliferation and ultimately reduce tumor recurrence [8,9,10,11]. The 2015 American Thyroid Association (ATA) guidelines recommend the maintenance of serum TSH levels at ≤2 mIU/L during the postoperative follow-up period in patients with low-risk PTC who underwent lobectomy [12]. However, there is insufficient evidence to determine whether maintaining postoperative TSH levels at ≤2 mIU/L is beneficial for reducing tumor recurrence in patients with low-risk PTC after lobectomy. A previous study reported that approximately 70% of patients after lobectomy had their TSH levels exceeding 2 mIU/L within 1-year after surgery [13], which means that a large proportion of patients require thyroid hormone supplementation to maintain their TSH levels ≤ 2 mIU/L even if their TSH level is within the upper normal range. This can lead to negative effects, such as impaired quality of life, increased medical costs, and side effects due to excessive thyroid hormone supplementation.

Previous studies have investigated the association between maintaining postoperative TSH levels at ≤2 mIU/L and tumor recurrence in patients with low-risk PTC after lobectomy; however, the results have been inconsistent [14,15,16]. An important drawback of previous studies is that although the TSH status of patients can fluctuate during the follow-up period, they only used the average or partial TSH levels to determine the postoperative TSH status of patients, which may insufficiently reflect the overall TSH status of patients during the follow-up period. Furthermore, these studies used a relatively short period of time to assess the TSH status of patients after surgery or until tumor recurrence after surgery, which can lead to inaccurate conclusions regarding the association between postoperative TSH status and tumor recurrence.

In this study, we investigated the association between maintaining the serum TSH levels at ≤2 mIU/L and tumor recurrence in patients with low-risk PTC who underwent lobectomy. To accurately determine the TSH status of patients during the follow-up period, we collected all the TSH test results during the 5-year postoperative period and calculated the ratio of test results that did not fulfill the target TSH levels according to the 2015 ATA guidelines. In addition, to investigate the effect of postoperative TSH status on tumor recurrence over a sufficient period, a 5-year landmark analysis was performed only for patients who had no recurrence for 5 years after the surgery.

## 2. Materials and Methods

### 2.1. Study Cohorts and Patient Inclusion

We retrospectively reviewed the medical records of 1165 thyroid cancer patients who underwent thyroid lobectomy at St. Vincent’s Hospital between January 2010 and December 2016. The medical records were examined for the clinicopathological factors of the patients, including the age, sex, tumor size, tumor location, multifocality, histologic subtype, lymphatic invasion, vascular invasion, extrathyroidal extension (ETE), the necessity to perform central lymph node dissection (CLND), lymph node metastasis status, and the results of thyroid function tests (TFT) at preoperative and postoperative follow-up period, which included the serum free thyroxine (fT4), triiodothyronine (T3), and TSH levels. The surgeon decided whether to perform CLND based on preoperative examination or intraoperative findings. If CLND was performed, only the side on which lobectomy was performed was included. To create a histologically homogenous patient population, we excluded the patients with the following: follicular thyroid carcinoma (*n* = 4), medullary thyroid carcinoma (*n* = 1), PTC with oncocytic variants (*n* = 2), and PTC with Warthin-like variants (*n* = 1). In order to select the patients who fulfilled the criteria of low-risk PTC according to the 2015 ATA guidelines, we excluded the patients with the following pathologic characteristics: gross ETE (*n* = 33), microscopic ETE (*n* = 29), lymph node metastasis with a size ≥ 0.2 cm or number > 5 (*n* = 36), and tall cell variant (*n* = 2). None of the patients exhibited vascular invasion. Patients who underwent completion thyroidectomy immediately after lobectomy (*n* = 29), those who did not undergo TFT testing at least once a year during the 5-year follow-up period after lobectomy (*n* = 109), and had no or inadequate clinicopathological data for analysis, including preoperative TFT (*n* = 13) and histological subtype (*n* = 3), were excluded. For the 5-year landmark analysis, we excluded patients who experienced tumor recurrence within 5 years after lobectomy (*n* = 28). Because our study aimed to include the patients with normal thyroid function preoperatively and TSH levels below the upper limit of the normal reference range during the 5-year follow-up period, we excluded the patients with abnormal preoperative TFT (*n* = 83) and those whose TSH level was higher than the upper normal range even once during the 5-year follow-up period (*n* = 129) based on the TFT reference ranges of our hospital (fT4: 0.6–1.3 ng/dL, T3: 0.87–1.78 ng/mL, TSH: 0.38–5.33 mIU/L). Finally, 662 patients were included in this study (Figure 1).

### 2.2. Definition of Serum TSH Status during Follow-Up

All patients started levothyroxine intake within 1 month after lobectomy and underwent levothyroxine dose adjustment during the postoperative follow-up period in the outpatient clinic. Physical examination, measurement of TFT, including fT4, T3, TSH, and imaging studies, including neck ultrasonography, chest radiography, and neck computed tomography (CT), were regularly performed every 6–12 months, and additional imaging studies, including chest CT, bone scan, or positron emission tomography with fluorodeoxyglucose/CT were performed if needed. During the 5-year follow-up after lobectomy, all patients included in this study had at least one serum TSH measurement per year; hence, all patients had a minimum of 5 and a maximum of 10 serum TSH measurements during the 5-year follow-up after lobectomy. The serum TSH status of patients during the follow-up period was determined based on the ‘TSH > 2 ratio’, which was defined as the number of times that the serum TSH was > 2 mIU/L during the 5-year follow-up period after lobectomy divided by the total number of TSH measurements during the 5-year follow-up period after lobectomy.

### 2.3. Outcome

The endpoint of interest was tumor recurrence. Recurrence-free survival (RFS) was defined as the time from surgery to tumor recurrence at any site or the last follow-up. To evaluate the prognostic effect of patients’ postoperative serum TSH status over a sufficient period, we set the landmark period to 5 years after lobectomy.

### 2.4. Statistical Analyses

Descriptive characteristics were summarized as numbers and percentages for categorical variables and as means ± standard deviations for continuous variables. The normality of the distribution of continuous variables was tested using the Shapiro–Wilk or Kolmogorov–Smirnov test, and variance equality was assessed using Levene’s test. Differences between the clinicopathological subgroups were compared using independent *t*-tests or Mann–Whitney tests for continuous variables and Pearson’s chi-square test or Fisher’s exact test for categorical variables. To determine the optimal cutoff value of the ‘TSH > 2 ratio’ during the 5-year follow-up period after lobectomy for risk estimation of tumor recurrence, a receiver operating characteristic (ROC) curve analysis was used, and area under the curve (AUC), sensitivity, and specificity were calculated. Survival curves were generated using the Kaplan–Meier method, and significance was determined using log-rank tests. The assumption of proportional hazard was checked using the Schoenfeld residual test, and *p* > 0.05 was considered to fulfill the assumption. Univariable and multivariable analyses were conducted using the Cox proportional hazards regression model for survival, with hazard ratios (HR) and 95% confidence intervals (CI) estimated for each variable. All statistical tests were 2-sided, and statistical significance was set at *p* < 0.05. All statistical analyses were performed using the SPSS Statistics for Windows (version 17.0; SPSS Inc., Chicago, IL, USA) or R version 4.2.2 (R Foundation for Statistical Computing, Vienna, Austria).

## 3. Results

### 3.1. Patients and Tumor Characteristics

Baseline characteristics of the study cohort are presented in Table 1. Of the 662 patients, the mean age was 48.29 ± 16.09 (range: 17–77) years, and 567 patients (85.6%) were female. The mean follow-up duration was 113.43 ± 30.19 (range: 60–167) months, and the mean number of TSH tests during the 5-year follow-up period after lobectomy was 9.23 ± 1.25 (range: 5–10) times. We performed the ROC curve analysis to determine the optimal cutoff value of the ‘TSH > 2 ratio’ during the 5-year follow-up period after lobectomy for predicting tumor recurrence, which showed that a ‘TSH > 2 ratio’ of 0.1833 during the 5-year follow-up period after lobectomy was the optimal cutoff value for predicting tumor recurrence, with an AUC of 0.687 (sensitivity: 51.9%, specificity: 76.4%; Figure 2). According to the ROC curve analysis results, patients were divided into two groups: ‘TSH > 2 ratio’ < 0.1833 (*n* = 498) and ‘TSH > 2 ratio’ > 0.1833 (*n* = 164). Table 1 compares the clinicopathological characteristics between the two groups. The patients in the group with a TSH > 2 ratio of > 0.1833 were significantly more likely to have a younger age (*p* = 0.03), higher proportion of tumor size > 1 cm (*p* = 0.009), lymphatic invasion (*p* = 0.002), less CLND (*p* < 0.001), higher number of TSH tests during the 5-year follow-up period after lobectomy (*p* = 0.01), and shorter follow-up duration (*p* < 0.001) than those of patients in the group with a TSH > 2 ratio of ≤ 0.1833. The mean ‘TSH > 2 ratio’ during the 5-year follow-up period after lobectomy was 0.1053 ± 0.1500 in all patients, which was higher in the ‘TSH > 2 ratio’ > 0.1833 (0.3265 ± 0.1334) group compared with the ‘TSH > 2 ratio’ ≤ 0.1833 (0.0324 ± 0.0510) group (*p* < 0.001).

### 3.2. Association between the ‘TSH > 2 Ratio’ during the 5-Year Follow-Up Period after Thyroid Lobectomy and Tumor Recurrence

After the 5-year landmark, tumor recurrence occurred in 27 patients (4.1%) who did not experience recurrence during the 5-year follow-up period after lobectomy (Table 1). The site of recurrence was the contralateral lobe in 18 patients (2.7%) and the regional lymph nodes in 9 patients (1.4%). None of the patients developed distant metastases during the follow-up period (Table 2). We investigated the differences in RFS between the two groups according to the ‘TSH > 2 ratio’ during the 5-year follow-up period after lobectomy (Figure 3). The Kaplan–Meier plot shows that patients with ‘TSH > 2 ratio’ > 0.1833 during the 5-year follow-up period after lobectomy had a worse RFS outcome than that of patients with ‘TSH > 2 ratio’ ≤ 0.1833 after the 5-year landmark (*p* < 0.001). We then conducted the univariable and multivariable analyses using the Cox proportional hazards regression model to determine the risk factors for tumor recurrence. Before running the Cox regression model, the assumption of proportional hazards was checked using the Schoenfeld residual test, and it was confirmed that the assumption of proportional hazards was fulfilled (*p*-value of the global test = 0.522). The results of the univariable and multivariable analyses are presented in Table 3. In the multivariable analysis, a ‘TSH > 2 ratio’ > 0.1833 during the 5-year follow-up period after lobectomy was found to be a significant risk factor for tumor recurrence after the 5-year landmark (HR: 4.795, 95% CI: 2.102–10.937, *p* < 0.001) along with a tumor size of >1 cm (HR: 3.080, 95% CI: 1.138–8.334, *p* = 0.027), presence of lymphatic invasion (HR: 7.542, 95% CI: 2.080–27.349, *p* = 0.002), and presence of multifocality (HR: 3.103, 95% CI: 1.354–7.113, *p* = 0.007).

## 4. Discussion

We investigated the association between TSH status during a 5-year postoperative follow-up period and tumor recurrence through a 5-year landmark analysis, considering the target TSH levels of the 2015 ATA guidelines in patients with low-risk PTC who underwent lobectomy. To effectively evaluate how well the target TSH level of the 2015 ATA guidelines was achieved in our study cohort, we calculated the ‘TSH > 2 ratio’ using all the TSH test results during the 5-year follow-up period after lobectomy and established an optimal cutoff value for predicting tumor recurrence. Our study results confirmed that patients with a ‘TSH > 2 ratio’ > 0.1833 during the postoperative 5-year follow-up period had worse RFS than that of patients with a ‘TSH > 2 ratio’ < 0.1833. This finding suggests that maintaining the TSH status of patients within the recommendation of the 2015 ATA guidelines above a certain percentage during the postoperative follow-up period is important for reducing the recurrence rate in low-risk PTC patients after lobectomy.

Since the 2015 ATA guidelines recommended keeping serum TSH levels ≤ 2 mIU/L to reduce tumor recurrence in patients with low-risk PTC who underwent lobectomy, there has been ongoing controversy about its necessity. Previous studies have reported the association between postoperative TSH level ≤ 2 mIU/L and tumor recurrence in patients with low-risk PTC after lobectomy, wherein a study conducted by Park et al. including 639 low-risk patients who underwent lobectomy with a median follow-up period of 8.6 years showed no difference in recurrence rate between patients with postoperative TSH level < 2 mIU/L and ≥2 mIU/L [15]. This study has the advantage of a relatively long follow-up duration and a propensity score-matched cohort. However, the homogeneity of study patients with PTC is low because 3–4% of the patients included in the analysis had follicular thyroid carcinoma. The TSH test results of patients during the follow-up period were not reflected in this study because the TSH status during the follow-up period was determined according to the TSH status obtained for more than 75% of their measurements during the follow-up period after lobectomy. A further study by Lee et al. reported that the postoperative 5-year TSH status did not affect tumor recurrence after lobectomy in patients with low-risk PTC, which included 1528 patients with a mean follow-up period of 5.6 years [16]. This study classified patients into four groups according to the postoperative 5-year TSH status (<0.5 mIU/L, 0.5–1.9 mIU/L, 2–4.4 mIU/L, and ≥4.5 mIU/L) and included only patients with PTC, which contributed to the high homogeneity of the study population. However, the TSH status of the patients was determined based on the average or dominant TSH value in the 5-year postoperative follow-up period, which is not representative of the TSH test results of the patients during the follow-up period. Furthermore, 34% and 13% of the study patients had ETE and lymph node metastasis, respectively, and did not meet the criteria for a low-risk group as defined by the 2015 ATA guidelines. In contrast, in Bae et al.’s study that included 369 patients with unilateral PTC with a median follow-up period of 72 months, only patients with postoperative TSH levels of > 2 mIU/L experienced tumor recurrence during the follow-up period after lobectomy. In this study, long-term follow-up information on the TSH status of patients after surgery was lacking because the TSH status was determined based on the TSH value at 1 year after lobectomy [9]. Recently, Won et al. conducted a meta-analysis to evaluate the effectiveness of TSH maintenance below 2 mIU/L in patients treated with thyroid lobectomy for low-risk differentiated thyroid carcinoma and concluded that the evidence for the effectiveness of postoperative TSH maintenance below 2 mIU/L is insufficient [17]. The authors pointed out the relatively high heterogeneity among the included studies (I^2^ value of 87.43%) and the possibility of TSH status fluctuations during the postoperative follow-up period as limitations of this study. Considering the limitations of previous studies, we only included patients who fulfilled the 2015 ATA guideline criteria for low-risk PTC through thorough patient selection, and we determined the TSH status of patients using the entire TSH test results during the 5-year follow-up period after lobectomy. Despite the uncertainty of the need for postoperative TSH maintenance ≤ 2 mIU/L in patients with low-risk PTC after lobectomy, a recent study reported that many clinicians consider TSH suppression even at levels below 0.5 or 0.1 mIU/L in patients with low-risk PTC or DTC in actual practice [18,19]. To resolve this discrepancy between guidelines and clinical practice, a prospective study is currently underway [20].

Although the 2015 ATA guidelines recommend maintaining postoperative TSH levels between 0.5 and 2 after lobectomy in patients with low-risk PTC, there is no recommendation on how long the TSH status should be maintained. There have been no studies on the optimal duration of postoperative TSH levels in patients with low-risk PTC after lobectomy. The 2022 National Institute for Health and Care Excellence guidelines only stated that lifelong TSH suppression after surgery was not necessary unless patients had high-risk or metastatic disease but did not suggest an optimal duration of TSH suppression [21]. Theoretically, thyroid cancer cells can proliferate during exposure to high TSH levels, and the degree of thyroid cancer cell proliferation may increase proportionally with longer exposure times. A previous study reported that 4 of 10 recurrences occurred in patients with low-risk PTC even after 5 years after the surgery, with an average follow-up period of 8.6 years [15]. In our study, among the 55 patients who experienced recurrence during the entire follow-up period after lobectomy, 27 experienced recurrence even 5 years after lobectomy. Such late recurrences have been reported frequently, probably due to the slow progression of low-risk PTC; therefore, previous studies have reported that TSH suppression therapy for at least 5–10 years after surgery is appropriate [22]. It may be reasonable to investigate the association between postoperative TSH status and tumor recurrence over a long follow-up period rather than a short follow-up period in patients with low-risk PTC. However, the TSH status of patients may fluctuate during a long follow-up period owing to variables such as noncompliance with levothyroxine intake, changes in absorption rate with age, and inconsistent dosing time. If an unintentional rise in TSH levels occurs despite the efforts of the clinician to maintain the TSH at target levels and the longer the period continues, it can have a negative impact on tumor recurrence. Therefore, considering the results of our study, clinicians should make efforts to maintain the appropriate TSH status of patients for a long period through a thorough examination during the postoperative follow-up period.

Some clinicians perform regular follow-up examinations at intervals of 6 months to 1 year for 5 years after lobectomy for patients with low-risk PTC and reduce the frequency and number of examinations thereafter if tumor recurrence is not detected at 5 years after lobectomy. According to a recent study, it seems reasonable to de-escalate the follow-up period for patients with low-risk PTC after lobectomy [23]. This is an acceptable strategy considering the indolent clinical course and low recurrence rate of low-risk PTC. However, based on our results, the probability of tumor recurrence can be significantly higher even after 5 years if the TSH status of the patient is not appropriately maintained during the postoperative 5-year follow-up period. Such patients may require careful long-term follow-up even after 5 years postoperatively. In addition, providing this information to patients can have a positive effect on maintaining patient compliance with long-term follow-up by raising awareness of their condition.

This study has several limitations. First, the data were retrospectively collected, indicating a potential selection bias. Second, although not insufficient compared to previous studies, the number of recurrent events in our study was relatively small, which may have negatively affected the power of the statistical analyses. Third, although previous studies have reported that BRAF mutations are a significant prognostic factor in patients with low-risk PTC [24,25], the current study did not evaluate the BRAF mutation status of patients because of a lack of records. Fourth, some patients in our study experienced TSH levels below 0.5 mIU/L during the follow-up period, although the 2015 ATA guidelines recommend a target TSH level of 0.5 to 2 mIU/L for patients with low-risk PTC after lobectomy. This may be because most patients in our study received levothyroxine after lobectomy, although the dose was later adjusted. However, it seems unlikely that these patients affected the results because there is currently no evidence that low TSH levels increase the recurrence rate in patients with low-risk PTC after lobectomy. Fifth, we did not conduct a more detailed subgroup analysis for each site of tumor recurrence, considering all possible clinicopathologic factors, including lymph node metastasis and multicentricity.

## 5. Conclusions

This study showed that maintaining an appropriate postoperative TSH level that meets the 2015 ATA guideline recommendations below a certain percentage of the total TSH results has a negative impact on tumor recurrence in patients with low-risk PTC after lobectomy. Clinicians need to make deliberate efforts to maintain the TSH status of patients within the appropriate range during the postoperative follow-up period, and long-term follow-up, even after 5 years after the surgery, may be considered if the TSH status of patients is not appropriately maintained during the 5-year follow-up period after lobectomy. Further studies are needed to evaluate the prognostic impact of the TSH status of patients and the optimal duration of TSH suppression in low-risk PTC after lobectomy.

## Figures and Tables

**Figure 1 cancers-16-03253-f001:**
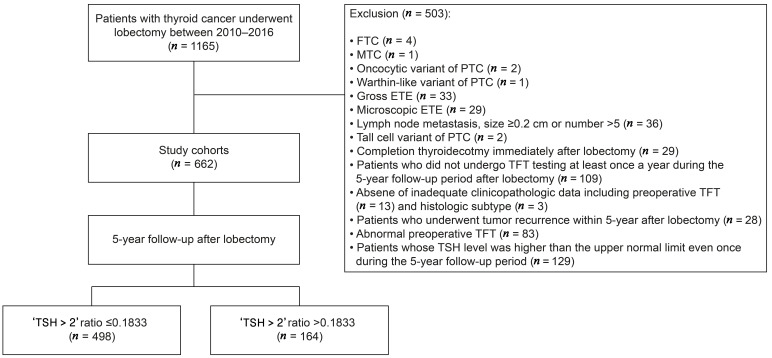
Flowchart of patient inclusion. FTC, follicular thyroid carcinoma; MTC, medullary thyroid carcinoma; PTC, papillary thyroid carcinoma; ETE, extrathyroidal extension; TFT, thyroid function test; TSH, thyroid-stimulating hormone.

**Figure 2 cancers-16-03253-f002:**
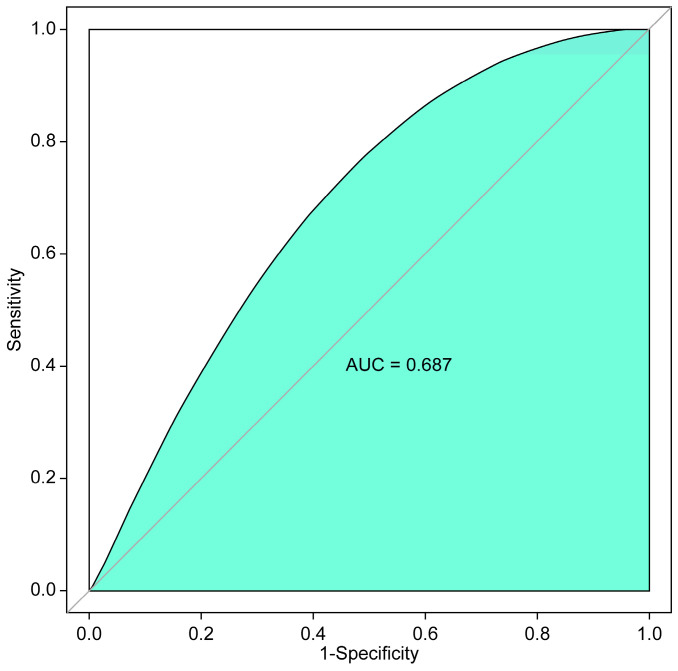
ROC curve analysis to determine the optimal cutoff value of the ‘TSH > 2 ratio’ during the 5-year follow-up period after lobectomy for predicting tumor recurrence. ROC, receiver operating characteristic; TSH, thyroid-stimulating hormone; AUC, area under the curve.

**Figure 3 cancers-16-03253-f003:**
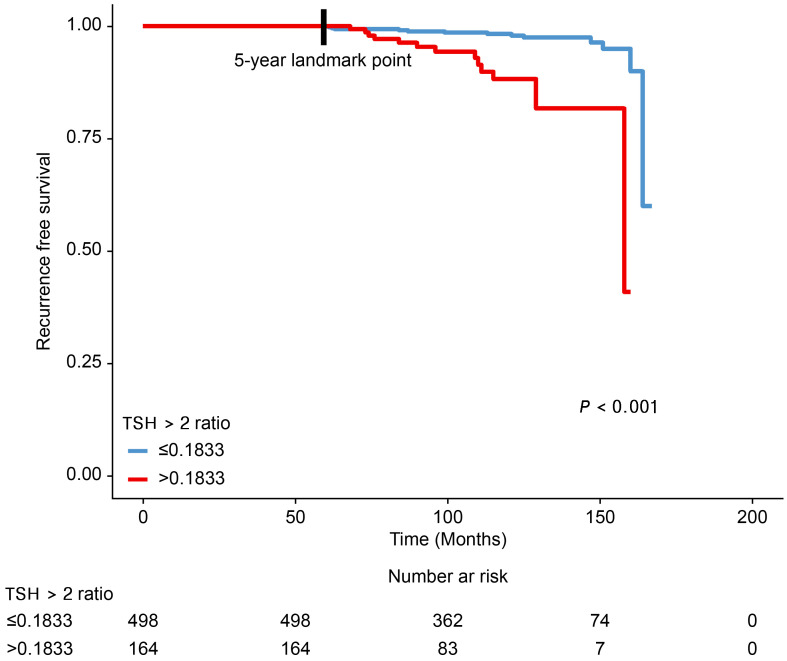
Comparison of the differences in the recurrence-free survival rates between patient groups according to the ‘TSH > 2 ratio’ during the 5-year follow-up period after lobectomy. TSH, thyroid-stimulating hormone.

**Table 1 cancers-16-03253-t001:** Baseline patient characteristics.

	All Patients	‘TSH > 2 Ratio’ ≤ 0.1833 (*n* = 498)	‘TSH > 2 Ratio’ > 0.1833 (*n* = 164)	*p*-Value
Age (years)	48.29 ± 16.09	48.99 ± 17.48	46.15 ± 10.61	0.03
≤55	513 (77.5%)	378 (75.9%)	135 (82.3%)	0.088
>55	149 (22.5%)	120 (24.1%)	29 (17.7%)	
Sex				
Female	567 (85.6%)	427 (85.7%)	140 (85.4%)	0.905
Male	95 (14.4%)	71 (14.3%)	24 (14.6%)	
Tumor size (cm)	0.64 ± 0.35	0.63 ± 0.31	0.67 ± 0.44	0.694
≤1 cm	609 (92.0%)	466 (93.6%)	143 (87.2%)	0.009
>1 cm	53 (8%)	32 (6.4%)	21 (12.8%)	
Tumor location				
Right	337 (50.9%)	252 (50.6%)	85 (51.8%)	0.657
Left	317 (47.9%)	241 (48.4%)	76 (46.3%)	
Isthmus	8 (1.2%)	5 (1.0%)	3 (1.8%)	
Multifocality				
No	566 (85.5%)	431 (86.5%)	135 (82.3%)	0.182
Yes	96 (14.5%)	67 (13.5%)	29 (17.7%)	
Histologic subtype				
Classic	623 (94.1%)	469 (94.2%)	154 (93.9%)	0.897
Follicular variant	39 (5.9%)	29 (5.8%)	10 (6.1%)	
Lymphatic invasion				
No	650 (98.2%)	494 (99.2%)	156 (95.1%)	0.002
Yes	12 (1.8%)	4 (0.8%)	8 (4.9%)	
Central lymph node dissection				
No	439 (66.3%)	311 (62.4%)	128 (78.0%)	<0.001
Yes	223 (33.7%)	187 (37.6%)	36 (22.0%)	
Number of TSH tests during the 5 years after lobectomy	9.23 ± 1.25	9.17 ± 1.27	9.4 ± 1.16	0.01
FU duration (months)	113.43 ± 30.19	117.28 ± 30.50	101.75 ± 26.02	<0.001
‘TSH > 2 ratio’ during the 5 years after lobectomy	0.1053 ± 0.1500	0.0324 ± 0.0510	0.3265 ± 0.1334	<0.001
Recurrence after the 5-year landmark point				
No	635 (95.9%)	485 (97.4%)	150 (91.5%)	0.001
Yes	27 (4.1%)	13 (2.6%)	14 (8.5%)	

TSH, thyroid stimulating hormone; FU, follow-up.

**Table 2 cancers-16-03253-t002:** Recurrence demographics.

	All Patients	‘TSH > 2 Ratio’ ≤ 0.1833 (*n* = 498)	‘TSH > 2 Ratio’ > 0.1833 (*n* = 164)
No recurrence	635 (95.9%)	485 (97.4%)	150 (91.5%)
Contralateral lobe	18 (2.7%)	8 (1.6%)	10 (6.1%)
Regional lymph nodes	9 (1.4%)	5 (1.0%)	4 (2.4%)
Distant metastases	0 (0%)	0 (0%)	0 (0%)

TSH, thyroid stimulating hormone.

**Table 3 cancers-16-03253-t003:** Univariable and multivariable Cox regression analyses for tumor recurrence.

Variable		Univariable		Multivariable ^1^	
		HR (95% CI)	*p*-Value	HR (95% CI)	*p*-Value
Age, years	≤55	1 (Reference)	–	–	–
	>55	1.498 (0.594–3.780)	0.392	–	–
Sex	Female	1 (Reference)	–	–	–
	Male	1.155 (0.437–3.052)	0.772	–	–
Tumor size	≤1 cm	1 (Reference)	–	1 (Reference)	–
	>1 cm	2.448 (0.835–7.173)	0.103	3.080 (1.138–8.334)	0.027
Tumor location	Rt.	1 (Reference)	–	–	–
	Lt.	0.505 (0.211–1.206)	0.124	–	–
	Isthmus	0.999 (0.107–9.331)	0.999	–	–
Histologic subtype	Classic	1 (Reference)	–	–	–
	Follicular variant	2.415 (0.671–8.689)	0.177	–	–
Lymphatic invasion	No	1 (Reference)	–	1 (Reference)	–
	Yes	8.118 (2.072–31.803)	0.003	7.542 (2.080–27.349)	0.002
Central lymph node dissection	No	1 (Reference)	–	–	–
	Yes	0.873 (0.373–2.042)	0.754	–	–
Multifocality	No	1 (Reference)	–	1 (Reference)	–
	Yes	3.439 (1.477–8.011)	0.004	3.103 (1.354–7.113)	0.007
TSH > 2 ratio	≤0.1833	1 (Reference)	–	1 (Reference)	–
	>0.1833	4.758 (2.033–11.135)	<0.001	4.795 (2.102–10.937)	<0.001

^1^ Adjusted for age, sex, tumor size, tumor location, histologic subtype, lymphatic invasion, central lymph node dissection, multifocality, and TSH > 2 ratio. TSH, thyroid stimulating hormone; HR, hazard ratio.

## Data Availability

The data presented in this study are available at the request of the corresponding author because they are not publicly available due to privacy or ethical restrictions.

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
