# Peer review of "Prognostic Implications of Maintaining the Target Thyroid-Stimulating Hormone Status Based on the 2015 American Thyroid Association Guidelines in Patients with Low-Risk Papillary Thyroid Carcinoma after Lobectomy: A 5-Year Landmark Analysis"

_cancers, 2024, doi:10.3390/cancers16193253_

Round 1

Reviewer 1 Report

Comments and Suggestions for Authors

Thanks to the Editor to give me the opportunity to revise this manuscript about a very interesting and current topic of research.

The authors aimed to investigate retrospectively the impact of TSH value more than 2 uI/ml in patients submitted to lobectomy for low-risk PTC. The merit of the paper is to consider not only the single TSH value at certain post-surgery times, but overall the avarage time spent by the patients with their TSH within the ATA 2015 guidelines target.

Having said that, I have some concerns about the study:

1) In the "Methods" section and in the "Figure 1", authors declared to have used a criterion of LN metastasis <2 cm to consider PTC as a low risk. This cut-off does not correspond to that recommended by the ATA, which instead states for a 0.2 cm. Therefore, it is very important that Authors clarify this point, because if it is not a "typo", it mandates to revise all the cohort selection.

2) Based on the baseline characteristics of the considered patients, those that fall within the "TSH>2 ratio > 0.1833" have statistically significant higher risk features in their path reports ( wider tumor, lymphatic invasion). It is important that authors make comment on it becuase the lower RFS detected in this gourp might be related to relative higher risk histology compared to the PTC found in the "TSH>2 ratio <0.1833" group.

3) The results about recurrence are not very clear. In fact in "Table 1" there is the rate of recurrence regarding the entire cohort and it is not commented on the "Results" section. On the other side, since the final sentence of the "Introduction" section, Authors state that they aimed to evaluate the rate of recurrence after a "5-years landmark". So, is the number of 27 (4.1%) collected during the 5 years post lobectomy follow-up or after that period of time (as anticipated by authors in the "Methods""Outcome subsection", where they wirte about 28 patients escluded for this subanalysis)? The "Figure 3" display the curves for the entire cohort of patients and not only for the subgroup of "no recurrence within 5 years post-lobectomy"

Author Response

Thanks to the Editor for giving me the opportunity to revise this manuscript about a very interesting and current topic of research.

The authors aimed to investigate retrospectively the impact of TSH value more than 2 uI/ml in patients submitted to lobectomy for low-risk PTC. The merit of the paper is to consider not only the single TSH value at certain post-surgery times, but overall the average time spent by the patients with their TSH within the ATA 2015 guidelines target.

Having said that, I have some concerns about the study:

Comment 1) In the "Methods" section and in the "Figure 1", authors declared to have used a criterion of LN metastasis <2 cm to consider PTC as a low risk. This cut-off does not correspond to that recommended by the ATA, which instead states for a 0.2 cm. Therefore, it is very important that Authors clarify this point, because if it is not a "typo", it mandates to revise all the cohort selection.

Response 1: Thank you for your thoughtful comment. Consistent with the reviewer’s comment, we note a typographical error in the “Methods” section and in “Figure 1.” We have corrected “<2cm” to “<0.2cm” and apologize for any confusion this error may have caused (Page 3, Line 105).

Comment 2) Based on the baseline characteristics of the considered patients, those that fall within the "TSH>2 ratio > 0.1833" have statistically significant higher risk features in their path reports (wider tumor, lymphatic invasion). It is important that authors make comment on it because the lower RFS detected in this group might be related to relative higher risk histology compared to the PTC found in the "TSH>2 ratio <0.1833" group.

Response 2: Thank you for your thoughtful comment, and we agree with the reviewer’s point. In Table 1, patients in the “TSH>2 ratio >0.1833” group exhibit more risk features, including tumor size and lymphatic invasion than those in “TSH>2 ratio ≤0.1833” group. However, as the reviewer is aware, the Kaplan–Meier graph for RFS in Figure 3 only accounts for tumor size (covariate) and tumor recurrence (event). Thus, the influence of other variables, such as tumor size and lymphatic invasion, is not considered, potentially biasing the RFS results. This is a common limitation of Kaplan–Meier analysis, not specifically mentioned in our manuscript. To mitigate such bias, we conducted a multivariable analysis (Table 2) to assess the impact of patients' TSH>2 ratio status on RFS, taking into account tumor size and lymphatic invasion. If desired, we will clarify this point further in the manuscript.

Comment 3) The results about recurrence are not very clear. In fact in "Table 1" there is the rate of recurrence regarding the entire cohort and it is not commented on the "Results" section. On the other side, since the final sentence of the "Introduction" section, Authors state that they aimed to evaluate the rate of recurrence after a "5-years landmark". So, is the number of 27 (4.1%) collected during the 5 years post lobectomy follow-up or after that period of time (as anticipated by authors in the "Methods" "Outcome subsection", where they write about 28 patients excluded for this sub-analysis)? The "Figure 3" display the curves for the entire cohort of patients and not only for the subgroup of "no recurrence within 5 years post-lobectomy"

Response 3: Thank you for your thoughtful comment; we concur with the reviewer’s observation. We apologize for any confusion resulting from unclear expressions. To mitigate this confusion, we have revised the term "recurrence" in Table 1 to "recurrence after the 5-year landmark point" and inserted "Table 1" in section 3.2, page 6, line 194.

- In “Figure 3”, the curves represent patients without recurrence within 5 years post-lobectomy. This likely stems from our omission of the 5-year landmark point on the graph. To prevent any misunderstanding, we have now included this landmark in Figure 3. Moreover, we previously noted in the 3.2 results section, page 6, lines 199–201, that patients with a ‘TSH>2 ratio’ >0.1833 during the 5-year follow-up after lobectomy exhibited poorer RFS outcomes than those of patients with a ‘TSH>2 ratio’ ≤0.1833 after the 5-year landmark.

Reviewer 2 Report

Comments and Suggestions for Authors

Good case number, and interesting study.

Two more things should be added

1) detail criteria of central lLN dissection:conditional or prophylactic.

Unilateral or bilateral dissection?

2)detail location of recurrence site and analysis with all possible associated pathological factors and clinical factors

Author Response

Good case number, and interesting study.

Two more things should be added

Comment 1) detail criteria of central lLN dissection:conditional or prophylactic.

Unilateral or bilateral dissection?

Response 1: Thank you for your thoughtful comment. We apologize for any confusion caused by our ambiguous phrasing. Although prior studies offer varied conclusions on the prognostic significance of prophylactic CLND in patients with low-risk papillary thyroid cancer, our study only involved unilateral (on the lesion side) therapeutic CLND, not prophylactic. We have clarified this distinction in the manuscript.

Comment 2) detail location of recurrence site and analysis with all possible associated pathological factors and clinical factors

Response: Thank you for your thoughtful comment, and we agree with the reviewer’s observation. We previously stated, “The site of recurrence was the contralateral lobe in 18 patients (2.7%) and the regional lymph nodes in 9 patients (1.4%). None of the patients developed distant metastases during the follow-up period.” in the 3.2 section (Page 6, Lines 195-197).

- We concur with the reviewer’s suggestion that a more detailed analysis at each recurrence site, considering all potential clinicopathologic factors, could yield more insightful conclusions. Nonetheless, the small number of recurrence events in our study precluded such a detailed subgroup analysis due to concerns about diminishing the statistical power. We have acknowledged this as a study limitation in the discussion section (Page 9-10, Lines 328-331).

Reviewer 3 Report

Comments and Suggestions for Authors

Very interesting work and addresses a point of great practical interest: whether or not to maintain low TSH.

I would question the inclusion/exclusion criteria. Despite using the ATA criteria, positive lymph nodes and multicentricity are already points of concern. I would suggest analyzing them separately.

In any case, even if this is not done, this would be a weak point in the work that should be discussed in more detail.

This suggestion in no way makes the work unfeasible; it would just be a way of pointing out points of doubt. Finally, it always seems strange to me to empty the central compartment with only a lobectomy. This point also deserves discussion.

Author Response

Very interesting work and addresses a point of great practical interest: whether or not to maintain low TSH.

Comment 1) I would question the inclusion/exclusion criteria. Despite using the ATA criteria, positive lymph nodes and multicentricity are already points of concern. I would suggest analyzing them separately.

In any case, even if this is not done, this would be a weak point in the work that should be discussed in more detail.

Response 1: Thank you for your thoughtful comment. Even though the metastatic lymph node status qualifies for the ATA low-risk category, we agree with the reviewer that minimal lymph node metastasis or tumor multicentricity could influence prognosis. We have documented this consideration as a study limitation in the discussion section (Page 9-10, Lines 328-331)..

Comment 2) This suggestion in no way makes the work unfeasible; it would just be a way of pointing out points of doubt. Finally, it always seems strange to me to empty the central compartment with only a lobectomy. This point also deserves discussion.

Response 2: Thank you for your thoughtful comment, and we agree with the reviewer’s point. While the prognostic significance of central compartment lymph node dissection in patients with low-risk PTC undergoing lobectomy alone remains uncertain, it is commonplace in South Korea for surgeons to perform either prophylactic or therapeutic central compartment lymph node dissections. In our study, only therapeutic unilateral central compartment lymph node dissections were performed based on preoperative or intraoperative findings, primarily because surgeons in Korea tend to confirm metastasis through such dissections without significantly increasing surgical complications. We have updated the “Methods” section to reflect the extent of CLND performed (Page 3, Lines 98-99). If desired, we will provide further details in the manuscript.

Round 2

Reviewer 1 Report

Comments and Suggestions for Authors

Thank the authors to addressed all my comments.

Now I think the paper is ready to be accepted by the Journal.

Author Response

Reviewer 1.

Thank the authors to addressed all my comments.

Now I think the paper is ready to be accepted by the Journal.

Response: Thank you for your thoughtful comment.

Reviewer 2 Report

Comments and Suggestions for Authors

Since recurrence is the primary observing point, authors can make a brief table similar table1 to describe those 27 recurrent  cases without statistical analysis.

Author Response

Reviewer 2.

Since recurrence is the primary observing point, authors can make a brief table similar table1 to describe those 27 recurrent cases without statistical analysis.

Response: Thank you for your thoughtful comment. We made a brief table as Table 2 (Page 6, lines 213–214) and described 27 recurrent cased without statistical analysis. (Previous Table 2 has been changed to Table 3.)  
